# Subgenomic RNAs and Their Encoded Proteins Contribute to the Rapid Duplication of SARS-CoV-2 and COVID-19 Progression

**DOI:** 10.3390/biom12111680

**Published:** 2022-11-12

**Authors:** Yifan Zhang, Xinglong Zhang, Huiwen Zheng, Longding Liu

**Affiliations:** Key Laboratory of Systemic Innovative Research on Virus Vaccine, Institute of Medical Biology, Chinese Academy of Medical Sciences and Peking Union Medical College, Kunming 650118, China

**Keywords:** SARS-CoV-2, subgenomic RNAs, replication, infection, immune evasion

## Abstract

Coronavirus disease 2019 (COVID-19), caused by severe acute respiratory syndrome coronavirus 2 (SARS-CoV-2), is currently widespread throughout the world, accompanied by a rising number of people infected and breakthrough infection of variants, which make the virus highly transmissible and replicable. A comprehensive understanding of the molecular virological events and induced immunological features during SARS-CoV-2 replication can provide reliable targets for vaccine and drug development. Among the potential targets, subgenomic RNAs and their encoded proteins involved in the life cycle of SARS-CoV-2 are extremely important in viral duplication and pathogenesis. Subgenomic RNAs employ a range of coping strategies to evade immune surveillance from replication to translation, which allows RNAs to synthesize quickly, encode structural proteins efficiently and complete the entire process of virus replication and assembly successfully. This review focuses on the characteristics and functions of SARS-CoV-2 subgenomic RNAs and their encoded proteins and explores in depth the role of subgenomic RNAs in the replication and infection of host cells to provide important clues to the mechanism of COVID-19 pathogenesis.

## 1. Introduction

Since the outbreak of the novel coronavirus (severe acute respiratory syndrome coronavirus 2 (SARS-CoV-2)) in 2019, public health has experienced enormous challenges. SARS-CoV-2, having high transmissibility and pathogenicity, can be transmitted stealthily via multiple routes [1,2,3,4]. Although the research and development of vaccines and drugs have brought a glimmer of light to COVID-19 prevention and treatment, the continuous mutation of SARS-CoV-2 epidemic strains has resulted in relevant etiological research lagging behind. Therefore, an in-depth understanding of the structural characteristics and pathogenic mechanism of SARS-CoV-2 is of great significance for controlling the virus. Replication of the viral genome within infected cells is a critical stage in the SARS-CoV-2 life cycle [5,6]. The production of a large number of progeny virions by rapid replication might trigger infiltration of inflammatory cells and release of inflammatory cytokines, which in severe cases can lead to acute lung injury and acute respiratory distress syndrome, and even death during infection [7,8,9]. The results of a study on 5-week-old golden hamsters infected with SARS-CoV-2 isolated from confirmed cases of COVID-19 indicated that peak virus titers were detected in the lungs from two days after virus infection until the seventh day [10]. In macaque infection models, the SARS-CoV-2 RNA loads of the oropharyngeal swabs reached up to 10^7^ copies/mL on the fifth day [11]. In clinical cases, the viral RNA load derived from respiratory samples peaked at 10^5^ copies/mL in the second week after the onset of symptoms, while it remained at 10^6^ copies/mL in the third week of severe symptoms [12]. In addition to upper and lower respiratory tract epithelial cells and lung tissues, some organs, such as the kidney, liver, and brain, also had low viral RNA levels [13]. The development of viral titers and clinical symptoms in different species and tissues mentioned above imply that SARS-CoV-2 invades the body for extensive replication and is not effectively detected by host antiviral immunity [14]. The SARS-CoV-2 mutant strains, especially the Omicron variant caused by the rapid global spread and massive recessive infection, exhibit enhanced virus replication and transmission. According to the genome sequence analysis of the variants, there are more than 60 substitutions, deletions, and insertions in the Omicron variant, which is the largest number of mutation sites among all SARS-CoV-2 variants thus far. Some mutations of the Omicron variant have been proven to be related to transmissibility, disease severity, and immune evasion [15,16], reflecting the unique replicative transmission and immune evasion capabilities of SARS-CoV-2.

## 2. Subgenomic RNAs Generation Mechanism and Functional Characteristics

The rapid replication of viruses is closely related to their genomic structural features and replication patterns. Negative-strand RNA virus ((−) strand RNA viruses) can be directly used as a template to form (+) strand RNA, which subsequently forms mRNA to synthesize proteins, such as influenza, with 8 viral RNA segments, which are translated into 12 proteins [17]. Positive-strand RNA viruses ((+) RNA viruses) usually generate genomic RNA by means of (−) strand RNA intermediates, but different viruses have their own replication strategies. Picornaviruses with smaller genomes, such as enterovirus EV71, use viral RNA as an mRNA template to encode polymer precursor proteins, which are cleaved into four structural and seven nonstructural proteins [18,19]. It is worth noting that coronaviruses take advantage of the subgenome replication strategy, which may be associated with easier interspecies transmission (Figure 1). SARS-CoV-2 viral genome structure consists of a genomic RNA (gRNA) and nine subgenomic RNAs (sgRNAs) [20,21]. The gRNA encodes 2 polyproteins, ORF1a and ORF1b, which can be translated into replicase polyprotein 1a (PP1a) and polyprotein 1ab (PP1ab) by host ribosomes and finally digested by viral protease into 16 nonstructural proteins (nsp1–16) [22,23,24]. Most of the nonstructural proteins constitute the replication transcription complex (RTC) involved in the regulation of viral replication and synthesis of subgenomic mRNA (sgmRNA) [25]; sgmRNAs encode four structural proteins including spike proteins (S), membrane proteins (M), envelope proteins (E), nucleocapsid proteins (N), and a variety of accessory proteins (ORF3a, ORF3b, ORF6, ORF7a, ORF7b, ORF8b, ORF9b, and ORF10), but the expressions of some accessory proteins have not been experimentally confirmed [26,27,28]. The subgenome replication process involves template switching, which requires the joint participation of the transcriptional-regulatory sequence (TRS) and replication-transcription complex (RTC) [5]. There are multiple TRSs in the SARS-CoV-2 genome, which are located upstream of 14 open reading frames, called the transcriptional-regulatory sequence-body (TRS-B). Similarly, the transcriptional-regulatory sequence-leader (TRS-L) is located in the 5′ untranslated region (5′UTR) [29,30]. The TRS-B can guide each ORF on the genome to the TRS-L, resulting in the formation of different subgenomic RNA fragments. RTC plays a central role in genomic replication and transcription, and the main active component is RNA-dependent RNA polymerase (RdRP). This subgenomic replication process is described below. When the RTC synthesizes (−) strand RNA intermediates along (+) strand RNA, template switching occurs if it encounters the TRS-B sequence in order to generate subgenomic RNA and then restarts adjacent to the TRS-L located at the 5′ end of the genome, leading to discontinuous transcription. The (−) strand RNA can be used as a template to synthesize (+) strand mRNA. Therefore, SARS-CoV-2 generates various subgenomic mRNAs through successive template-switching events, all of which carry the leader sequence 5′UTR and common 3′UTR [31,32]. Combining DNA nanoball sequencing and nanopore direct RNA sequencing, 7–9 viral RNAs can be detected in the cytoplasm of host cells infected by SARS-CoV-2. Of these, the largest sequence is a genomic RNA, and the rest are subgenomic RNAs and noncanonical subgenomic RNAs because of the noncanonical junctions between the 5′ leader sequence and downstream regions of the SARS-CoV-2 genome [33,34,35]. Template-switching events occur from (+) strand genomic RNA to (−) strand subgenomic RNAs in classic models of duplication. The latest research demonstrates that it can take place from (−) strand subgenomic RNAs to (−) strand subgenomic mRNA. Subgenomic RNAs can cascade multiple template transitions during production to generate shorter products [32], which heavily relies on long-distance RNA–RNA interactions to adjust the viral transcription and replication pathways [36], which is related to RNA–RNA pairing energy around template switching sites, consecutive pairing length, and especially the status of pair-end length [37]. Likewise, the RNA around template-switching sites utilizes secondary structures that bring TRS-B and TRS-L into physical proximity, which can increase the frequency of long-distance viral recombination in coronavirus [37]. This process of discontinuous transcription occurs in double-membrane vesicles (DMVs), which are double-membrane structures formed by the endoplasmic reticulum membrane. Research has shown that a large amount of accumulated double-stranded RNA is detected in double-membrane vesicles, which may be an intermediate product of viral genomic RNA replication and subgenomic mRNA synthesis [20,38,39].

The subgenome produces multiple RNAs in the abovementioned discontinuous transcription manner and translates them into structural proteins and accessory proteins simultaneously, accelerating the synthesis of viral replication complexes, which is the survival strategy of SARS-CoV-2 that allows it to escape immune surveillance and synthesize viral proteins efficiently. The sequence features of different subgenomic RNAs include 5’ untranslated region (5′UTR), protein-coding frame, and 3’ untranslated region (3′UTR). Among them, the 5′UTR contains a 75 nt leader sequence same as 1–75 nt of the gRNA 5′UTR, and a variable length RNA fragment (0–190nt). The 75 nt leader sequence contains 6–8 nucleotide core sequences (CS). It has been reported that the core leader sequence of SARS-CoV and SARS-CoV-2 is ACGAAC [40,41]. In coronaviruses, core sequences-leader (CS-L) can base pair with the nascent negative strand complementary to each core sequences-body (CS-B), a process required to drive template switching in subgenomes [24]. The variable RNA lengths of S-sgRNA, ORF3a-sgRNA, E-sgRNA, and N-sgRNA are 1–8nt, while M-sgRNA, 6-sgRNA, and 7b-sgRNA are 119, 230, and 151nt, respectively, resulting in 5′UTR of a specific length for different subgenomic RNAs [42]. In addition, there are some RNA elements in the UTR. The gRNA contains five stem-loop structures (SL1-SL5) in the 5′ UTR [43,44]. The leader sequences of subgenomic RNAs involve SL1-SL3. Studies have shown that stem-loop 1 (SL1) in the 5′ UTR leader of SARS-CoV and SARS-CoV-2 protects the virus from Nsp1-mediated repression of mRNA translation. The RNA structure in the UTR can also effectively regulate translation efficiency. For example, enterovirus EV71 is translated into a single polyprotein through an internal ribosome entry site (IRES) in the 5′UTR of the genome in a cap-independent way [45]. IRES, a cis-acting element, can recruit ribosomes used for host translation to internal viral RNA sites to initiate translation and regulate the viral replication cycle [46,47]. This RNA structure is found in a variety of pathogenic viruses, such as hepatitis A virus [48], hepatitis C virus (HCV) [49,50] and human immunodeficiency virus (HIV) [51,52]. Unlike the translation way of EV71, SARS-CoV-2 subgenomic mRNAs are translated in a cap-dependent manner that involves cis-acting RNA elements, such as the stem-loop structure of the leader sequence, the length of the poly(A) tail, and the role of elements [42]. Likewise, RNA structural elements in the 3′ UTR play a key role in the life cycle of viruses. For example, the stem-loop 2 motif (s2m) has a high degree of sequence conservation and exists in a variety of coronaviruses, including the 3′UTR in subgenome transcripts of SARS-CoV-2, and is easily affected by antisense oligonucleotides (ASO) targeting, is an ideal target for antiviral therapy [53].

## 3. The Effect of Subgenome Composition and Encoded Proteins on the Replication Ability of Virus Variants

A new mutation occurs every 10,000 nucleotides during RNA virus replication. The mutation rate of RNA viruses is 1,000,000 times higher than that of DNA viruses [54]. SARS-CoV-2 undergoes adaptive evolution caused by multiple factors, including nucleotide mutations and vaccine-induced immune stress during virus replication. The main drivers of SARS-CoV-2 variants lie in nucleotide sequence mutations, probably due to single-nucleotide polymorphisms (SNPs), insertions or deletions (INDELs) caused by discontinuous subgenomic RNA synthesis, and RNA modifications or editing driven by potential host factors, resulting in changes in viral replication capability, transmissibility, pathogenicity, and antigenicity [55]. (Figure 2) For example, single-nucleotide mutation sites of the Omicron variant are mainly located in the spike protein. There are approximately 32 mutation sites in this variant, including 30 base substitutions, 6 base deletions, and 3 base insertions [56], which increase the risk of reinfection and tolerance to vaccines. It has been reported that the complex encoded by nsp10–14 can remove misincorporated nucleotides or nucleotide analogs to improve the fidelity of RNA synthesis, proofreading the mismatched bases during RNA synthesis. Nsp14 is a bifunctional enzyme composed of an exonuclease (ExoN) domain and a methyltransferase (MTase) domain, which plays an important role in the replication and translation of SARS-CoV-2 [57]. For example, SARS-CoV-2 can affect the activity of nucleotide analog inhibitor antiviral drug remdesivir, which inhibits RdRP and significantly blocks coronavirus replication through nsp14 proofreading. RNA-dependent RNA polymerase (RdRP) is a key component of viral replication and transcription. Nucleotide incorporation errors mediated by RdRP mainly exist in the formation of negative strand genomic RNA (-gRNA) and subgenome RNAs (-sgRNAs), followed by the incorporation of erroneous bases into positive-strand genomic RNA (+gRNA) and subgenomic mRNAs (+sgRNAs), resulting in “error mutation” [58]. Subgenomic discontinuous transcription can easily cause high reorganization, resulting in the deletion of viral sequences or insertion of nonviral sequences to form defective interfering RNAs, which may alter the pathogenicity and virulence of the virus. For instance, coronaviruses HCoV-OC43 and HKU1 acquire the hemagglutinin esterase (HE) gene after recombination between precursor coronaviruses and influenza C-like viruses, which leads to loss of their sialic acid-binding activity through progressive deletions in their lectin domains [59,60]. A SARS coronavirus-attenuated vaccine lacking the full-length E gene can lead to partial replication of the E gene or insertion of a new sequence into the ORF8 gene after serial passage in vivo to improve virus fitness [61]. In addition, RNA modifications can modulate the RNA viral life cycle and affect the capacity of viral replication. For instance, the deletion of methylated m6A and Tl3/14 restricts HCV infectious particle production and protein expression in hepatitis C virus (HCV) [62,63]. Similarly, the deletion of METTL3 or the cytoplasmic m6A-related proteins YTHDF1 and YTHDF3 inhibits the replication of SARS-CoV-2 and HCoV-OC4359 [64].

Mutant strains can increase the viral transmission rate and risk of reinfection, reduce the protection provided by monoclonal neutralizing antibodies and vaccines [65], and enable SARS-CoV-2 to maintain or improve replication fitness [6], allowing it to continue to spread under the conditions of continuously growing herd immunity. The spike protein encoded by subgenomic RNA, a major mutation site in most variants during replication [66], mediates interactions with viruses and their receptors, which has a potential impact on the spread of viruses and immune evasion [67]. Among mutants, the D614G mutant strain had the most significant sequence variation, with a 56% variability rate [68]. Compared with the original strain, the infectivity of D614G increased by 4–9 times, accompanied by greater host binding capabilities and faster replication speeds [69,70]. Quantitative PCR results show that viral RNAs increase approximately threefold in patients infected with the D614 variant [71] and indicate that a higher proportion of functional spike protein may be why replicated viruses are more infectious [72]. Compared with the D614 variant, the Alpha, Beta, Gamma, and Delta variants have stronger replication and spread capabilities. Research indicates that viral loads of Delta variant-infected patients were 1260 times higher than viral loads of original strain-infected patients [73]. The site mutation caused by variants also influences the affinity between viruses and hosts and the level of neutralizing antibodies. For example, the N501Y mutation of the British mutant strain Alpha can directly affect the binding of the virus to host cells [74]. Other than the N501Y mutation, the South African mutant strain Beta and the Brazilian mutant strains Gamma (γ) and Zeta (ζ) have added E484K mutations, which can reduce the potency of neutralizing antibodies by up to 10 times in treatment-recovered patients [75]. Specific site mutations from different variants are the primary reason for the change in virus properties.

## 4. Subgenomic RNAs and Their Encoded Proteins Promote Immune Evasion of Viral Particles

SARS-CoV-2 has evolved a variety of strategies to escape host immune responses and increase the duration of infection and replication in vivo. Subgenomic RNA and its encoding protein are essential in promoting SARS-CoV-2 immune escape. At the RNAs level, shorter subgenomic RNAs can be generated by the above discontinuous transcription process effectively and escape recognition by host viral RNA sensors compared with genomic RNAs with a length of 30 kb. In addition, SARS-CoV-2 has a strict protective mechanism against modification of the replication environment and its own structure. The replication environment is composed of characteristic perinuclear double-membrane vesicles that provide a protective microenvironment for genomic RNA replication and mRNA transcription and avoid intermediate dsRNAs recognized by innate immune sensors [22,76]. More importantly, at the protein level, The SARS-CoV-2 subgenome encodes a variety of proteins that help viral particles evade innate and adaptive immune responses. Studies have shown that viral transcripts are expressed at high levels during viral infection, which allows the translation machinery within the host cell to be dominated by the production of viral proteins rather than host proteins [29]. The percentage of virus-encoded proteins among total cellular protein translation can increase 20,000-fold, and the ratio of the virus to cellular RNA can reach 90%, mostly subgenomic RNAs within 1–5 h of beta-coronavirus infection of cells [77]. The translation process in the early stages of infection requires the hijacking of host metabolites such as glucose and folic acid to meet the replication requirements for large-scale production of genomic RNA and highly abundant subgenomic RNAs [78]. In addition, the significantly increased expression of transcribed subgenomic RNAs during virus infection may be closely related to the pathogenesis of SARS-CoV-2.

Proteins encoded by subgenomic RNAs play different roles during viral replication. Among these coding proteins, structural proteins mainly complete the assembly of complete virus particles and play the role of virus pathogenicity. For example, nucleocapsid proteins can recognize and package genomic RNA into the ribonucleoprotein (RNP) complex even if detected by the host immune system during infection [79,80]. The spike and envelope proteins are putative virulence determinants that influence the assembly and release of viruses [81]. The membrane protein is a structural protein that is abundantly expressed in lipid membranes and is important for virus morphogenesis and interferon inhibition [82]. Unlike structural proteins, accessory proteins play an indispensable role in both innate and adaptive immune response evasion (Figure 3). The RIG-I-MAVS signaling pathway is considered to be the primary cytoplasmic RNA surveillance and protection system in innate immunity against viral infection. The pathway process is to recognize viral RNA through RIG-I-like receptors (RLRs), such as RIG-I, MDA5, and LGP2, activating innate immune signals and recruiting the mitochondrial outer membrane MAVS protein, which acts as a core adaptor protein for RLR signaling and controls downstream signaling. This signaling cascade culminates in the phosphorylation and activation of IRF3, IRF7, NF-κB, and AP-1 and the release of interferons and proinflammatory cytokines [83], which can bind to interferon receptors and initiate the JAK/STAT signaling cascade of pathways that translocate activated transcription factors to the nucleus to induce antiviral immune responses [84]. Innate immunity plays a crucial role in the clearance of foreign pathogens and induces an effective adaptive immune response. It has been reported that virus-infected patients with innate immune deficiency experience early viral replication in the upper respiratory tract and lungs and fail to initiate an adaptive immune response [85]. Therefore, evasion of innate immunity-mediated antiviral signals is a common defense strategy of pathogenic viruses in host replication and spread. As an RNA virus, the typical feature of SARS-CoV-2 immune evasion is the inhibition of type I and type III interferon (IFN)-mediated antiviral immunity [82]. The accessory proteins encoded by the subgenome are involved in many aspects of the above natural immune pathways. For example, SARS-CoV-2 variant Alpha (B.1.1.7) with stronger replication ability compared with the Wuhan strain, obviously increased subgenomic RNAs and protein levels of N, ORF9b, and ORF6, which are known innate immune antagonists [86]. Overexpressed ORF9b in the mitochondria can suppress innate immune responses by interacting with TOM70, which is a mitochondrial protein required to activate the adapter MAVS for sensing RNA [87]. ORF6 localizes to the nuclear pore complex (NPC) and directly interacts with Nup98-Rae1 through its C-terminal domain, which inhibits STAT1 and STAT2 nuclear translocation and attenuates the transcription and induction of IFN-stimulated gene-ISG, leading to IFN production and signaling conduction blockade [88]. Additionally, ORF10 induces autophagy to degrade MAVS expression and promote viral replication. In addition to performing important functions in innate immune responses, accessory proteins can also be involved in adaptive immunity mediated by antigen-presenting cells [89]. Host cells can present viral proteins to CD8+ T cells that can differentiate into cytotoxic T lymphocytes (CTLs) through histocompatibility complex-I. The surface T-cell receptors on CTLs recognize antigenic signals presented by the MHC-I-peptide complex and release perforin and granzyme, which directly induce the death of virus-infected cells and the production of cytokines such as interferon-γ, TNF-α, and IL-2 [90]. The ORF8 protein has been shown to promote the autophagic degradation of MHC in infected cells and disrupt the antigen presentation system, thus evading cellular immunity mediated by cytotoxic CD8+ T lymphocytes [91]. The histocompatibility complex MHC-II can form complexes with viral proteins and present them to CD4+ T cells. In 2019-nCoV patients who were 70% and 100% recovered, the immunological examination found that the CD4+ T-cell response to the S protein was strong and correlated with the anti-SARS-CoV-2 IgG and IgA titers. Statistical analysis found that M, S, and N proteins accounted for 11–27% of the total CD4^+^ response, with other responses typically targeting nsp3, nsp4, ORF3a, ORF8, and others [92]. Because of the effective innate immune evasion mechanism of SARS-CoV-2, the early CD4+ and CD8+ T-cell-induced adaptive responses may fail to play a protective role, and inflammation is further aggravated in the late stage.

## 5. Designing Vaccines and Drug Targets for Subgenomes

There are differences in the ease of transmission, the severity of related diseases, and the effectiveness of vaccines and therapeutics among SARS-CoV-2 variants, resulting in a lack of specific drugs for the treatment of COVID-19. Subgenomic RNAs are an important link in the replication life cycle of SARS-CoV-2. Designing relevant vaccines or drugs aimed at key steps in subgenomic engagement can directly or indirectly affect the replication and viability of the virus and the degree of the body’s immune response. According to the above biological characteristics and functional analysis, we believe that multiple stages of subgenome involvement can be used as potential targets for antiviral drugs, which are divided into several aspects. The first is the discontinuous transcriptional synthesis step of the subgenome, which is essential for the formation of a new generation of viral particles. The development of antiviral drugs based on this stage can fundamentally limit the extensive replication of the virus. RdRP is a key enzyme in the viral life cycle, not only for replication of the viral genome but also for discontinuous transcription of the subgenome [22]. In fact, the current updated drug research about the viral replication cycle focuses on the process of viral entry into cells and polymerase inhibitors. Whether it is a DNA virus or an RNA virus, the polymerase is a suitable target for blocking viral replication. Currently, many antiviral drugs contain RdRP-related enzyme inhibitors and are undergoing clinical trials, such as arbidol, remdisvir, favipiravir, EIDD-2081, and ribavirin [93]. Of these, remdisvir (RDV, originally developed to combat Ebola virus infection) is the only drug recently authorized to be used in hospitalized patients with coronavirus disease, showing good antiviral activity [94]. Therapeutic drugs targeting the highly conserved leader sequence of the 5′UTR of the subgenome can significantly reduce viral gene expression and replication. The study found that targeting the leader sequence of SARS-CoV-1 in the viral genome and subgenomic RNA can effectively inhibit the expression of viral genes (S, E, M, and N) and ultimately inhibit virus replication in Vero E6 cells [95]. Therefore, the SARS-CoV-2 5′UTR leader sequence is a potential indicator. At the immune response level, DMVs of the subgenomic replication environment are also potential breakthrough points [39,96]. They are the primary site for replication and transcription after coronaviruses infect cells, and disruption of this structure may expose the location of the virus and induce a host immune response. Studies have confirmed that Oxy210 and Oxy232, semisynthetic oxysterols, show strong anti-SARS-CoV-2 activity, disrupting DMV formation and reducing SARS-CoV-2 replication [97]. In addition, inhibition of subgenomic RNAs and their encoded proteins can affect the host’s innate and adaptive immune responses. For example, SARS-CoV-2 ORF8 is the only protein with approximately 20% homology to SARS-CoV, which can destroy the antigen presentation system and assist in virus immune evasion. The development of compounds that specifically target ORF8 and damage MHC-I antigen presentation can enhance the immune surveillance of SARS-CoV-2 infection [91]. At the metabolic level, host metabolism is very important for virion production, especially at the subgenomic RNA expression level. Therefore, host-based metabolically targeted therapies may provide targets for blocking viral replication. Antifolates, including methotrexate and SHMT inhibitors, have been approved for the treatment of COVID-19 [98]. In addition to affecting the purine synthesis pathway, antifolates can also act synergistically with the antiviral nucleotide analog remdesivir [99,100].

## 6. Conclusions

The pathogenic mechanism of SARS-CoV-2 is complicated and changeable, remaining unclear. We analyzed the structural features and infection conditions of the SARS-CoV-2 genome and found that there is an inseparable relationship between the rapid replication of the virus to assemble complete viral particles during infection and the functional characteristics of the subgenome. Subgenomic RNAs and their encoded proteins are present through the entire life cycle of viruses, from infection to replication and release, playing a guiding role in virus synthesis. In the process of viral replication, the genome generates multiple subgenomic RNAs through the discontinuous transcription mechanism mediated by transcriptional regulatory sequences and encodes structural proteins to assemble progeny virions that form mature virions, which not only speeds up synthesis of the subgenomic RNAs and increases translation efficiency but also provides the raw materials for viral assembly. Importantly, SARS-CoV-2 viral subgenomic RNAs create favorable conditions for protecting the virus from host immune responses in multiple ways during replication, including sequence information modification to recognize self-RNA, double-membrane vesicles to unpack RNA, and the subgenome to antagonize innate and adaptive immune responses at the RNA and protein levels. In addition, the subgenome hijacks host metabolism to support its own translation and synthesis in the process of replication and packaging. In conclusion, we believe that the subgenome is an important aspect of the rapid replication of SARS-CoV-2, but there are still many problems that have not yet been explained, such as the specific formation mechanism of double-membrane vesicles on the endoplasmic reticulum, the sequence differences between the subgenomes of the variant strains, the related operation mode of transcriptional regulatory sequences, and the significance of the common leader sequence of the subgenomes. These questions may provide useful information for subsequent vaccine design sequences and targets for inducing the body’s immune response and indicate the future direction of our work.

## Figures and Tables

**Figure 1 biomolecules-12-01680-f001:**
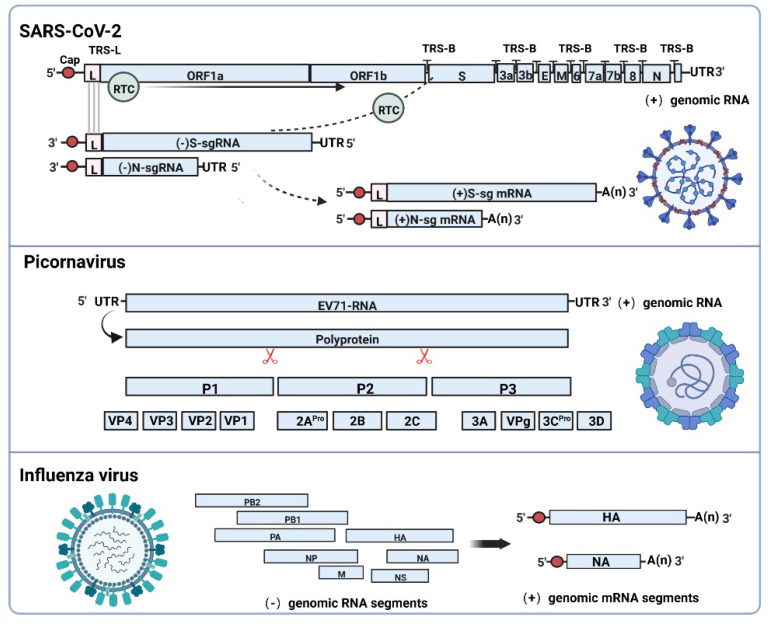
Comparison of transcription patterns of positive or negative strand viruses possessing genomes of different lengths. SARS-CoV-2 ((+) strand RNA virus) utilizes RTC (RdRP-formed complex) and TRS for discontinuous transcription to generate (−) strand subgenomic RNAs, which are used as translation templates for the synthesis of (+) strand subgenomic mRNAs. Picornaviruses with smaller genomes, such as enterovirus EV71, directly translate genomic RNA into multimeric proteins, which are cleaved into structural and nonstructural proteins. Influenza virus ((−) strand RNA virus) use itself as a template to synthesize the positive strand, which is transcribed into mRNA. Image created with BioRender (https://biorender.com/ (accessed on 8 August 2022)).

**Figure 2 biomolecules-12-01680-f002:**
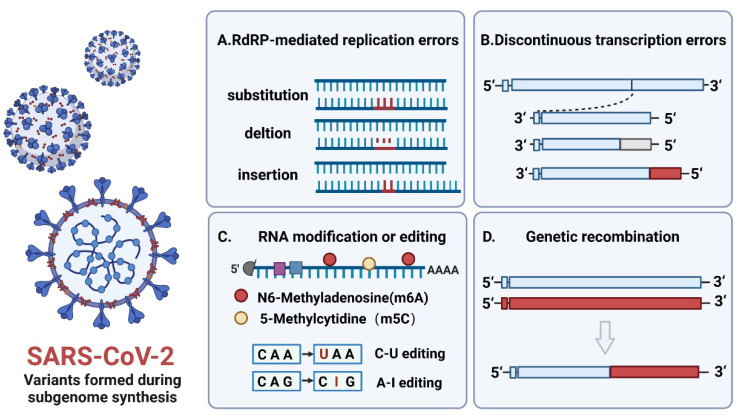
Potential mechanism of SARS-CoV-2 variants caused by sgRNAs during replication. (**A**) The sgRNAs sequence errors occur due to RdRP-mediated base substitution, base deletions, or base insertions. (**B**) Discontinuous transcription leads to defective sgRNAs formation, including loss of partial viral sequences (gray part) or addition of nonviral sequences (red part). (**C**) SgRNA replication properties alter because of host-mediated RNA modification or editing. (**D**) Genetic recombination may occur among different subgenomes to form new subgenomic types. The blue and red parts represent different subgenomic RNA, respectively. Image created with BioRender. (https://biorender.com/ (accessed on 8 August 2022)).

**Figure 3 biomolecules-12-01680-f003:**
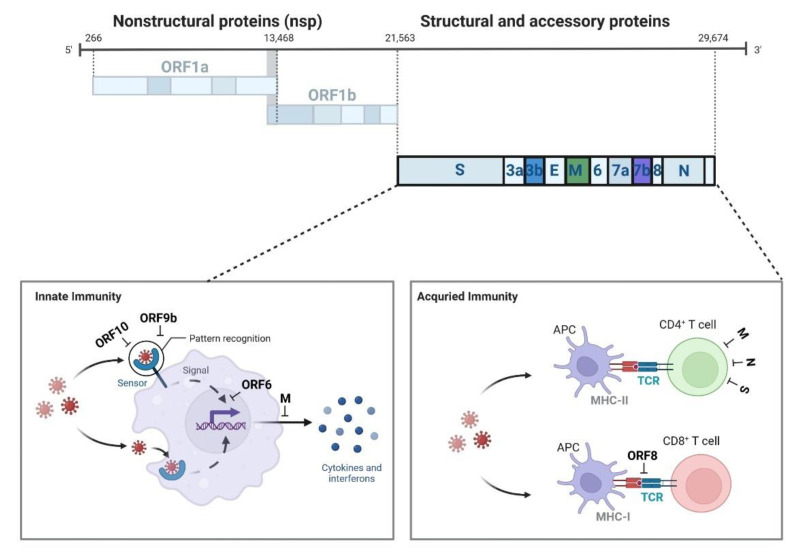
SARS-CoV-2 viral proteins and immune responses. SARS-CoV-2 genomic RNA encodes nonstructural proteins, and subgenomic RNA encodes structural and accessory proteins. Among them, subgenomic encoded proteins are involved in multiple steps of inhibiting the production and secretion of IFN-γ and antigen presentation, antagonizing the host innate and acquired immune responses. Image created with BioRender (https://biorender.com/ (accessed on 8 August 2022)).

## Data Availability

Not applicable.

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
