# Peer review of "Subgenomic RNAs and Their Encoded Proteins Contribute to the Rapid Duplication of SARS-CoV-2 and COVID-19 Progression"

_biomolecules, 2022, doi:10.3390/biom12111680_

Round 1
Reviewer 1 Report (Previous Reviewer 2)
The authors have added some descriptions which improved the manuscript. Some questions remain:
1. Still the manuscript seems to talk more about the proteins encoded by the subgenomic RNA. Suggest to change the title to "Subgenomic RNAs and their encoded proteins...". Otherwise, the manuscript reads very weak on the RNA side.
2. The change made on line 88 did not address my original question. The RTC not always begins to transcribe subgenomic RNA if it encounters the TRS-B. Because if this is the case, no full-length RNA will be produced.
3. Refer to my original question 2, I still did not get why the authors gave a detailed description about how discontinuous transcription occurs in section 4 but did not talk about this in section 2 where the mechanistic details should be included.
4. Please revise section-3 and section-4 substantially to focus on the subgenomic RNAs and their encoded proteins. The current manuscript reads like a collection of random pieces of information. For example, RdRP-mediated replication errors and RNA modifications are not unique aspects of subgenomic RNA. 5´-capping to evade immune response is also not unique to subgenomic RNA and thus should not be considered as an evidence of subgenomic RNA to promote immune evasion.
Author Response
Dear Reviewer,
Thank you for your comments on this manuscript. Please see the annex for specific modifications.
Best regards
Sincerely yours

Reviewer 2 Report (Previous Reviewer 3)
I believe the manuscript has improved considerably and is now ready for publication in biomolecules.
Author Response
Dear Reviewer,
We thank the referee again for taking the time to review our manuscript and your guidance and constructive comments on my manuscript. We have noticed many details and learned more knowledge in the process of revising the manuscript. Thank you for giving us the opportunity to publish.
Best regards
Sincerely yours
This manuscript is a resubmission of an earlier submission. The following is a list of the peer review reports and author responses from that submission.
Round 1
Reviewer 1 Report
Authors of the review “Subgenomic RNAs contribute to the rapid duplication of SARS-CoV-2 and COVID-19 progression” inform in the abstract that “This review focuses on the characteristics and functions of SARS-CoV-2 subgenomic RNAs and explores in depth the role of subgenomic RNAs in the replication and infection of host cells to provide important clues to the mechanism of COVID-19 pathogenesis.” However, the review provides very general information about subgenomic RNAs (sgRNAs), moreover presented reports are often incomplete and chaotically presented. The review is focused in large part on SARS-CoV-2 proteins (which are a separate subject) and related to proteins immune response. I did not evaluate correctness of this part. The review is less informative than already published papers regarding sgRNAs, e.g.,
DOI: 10.3390/v13101923
DOI: 10.1038/s41580-021-00432-z
I have impression that thorough research of literature regarding SARS-Cov-2 sgRNAs was neglected. For example, the authors report that a role of the leader sequence in genomic RNA (gRNA) and sgRNA is unknown. This is untrue, unprecise at best. Of course, there is a lot to learn about function of SARS-CoV-2 UTRs, however, there are publications indicating that the first stem-loop (SL1) is required for viral escape of NSP1-mediated translation inhibition (viral nsp1 protein mediates a translation shut-off for host mRNA). SL1 is promising target for antiviral therapy. Detailed construction of sgRNA is not clearly presented (leader sequence 75 nt plus 5’UTR sequences downstream of TRS-B; coding region and 3’UTR - sgRNA are polycistronic but functionally monocistronic [ORF most proximal to the 5′ end undergos translation] except in the case of sgRNA leaky scanning). RNA structural motifs determined in UTRs and the coding sequences of gRNA and sgRNAs are not mentioned in this review, while RNA structures have a large regulatory potential and may be targeted by inhibitors. The ‘non-canonical’ subgenomic transcripts were not discussed. Authors speculate that SARS-CoV-2 may use internal ribosome entry site (IRES, referring to EV71 and HCV viruses) but they do not comment literature reporting that initiation of translation occurs by cap-dependent mechanism.
Literature on Nsp1 – sl1, e.g.
DOI: 10.1261/rna.079086.121
DOI: 10.1101/2020.09.18.302901
DOI: 10.1073/pnas.2117198119
cap-dependent mechanism of translation
DOI: 10.1016/j.cell.2020.11.016
DOI: 10.3390/v14071505
DOI: 10.1038/s41467-020-17496-8
Figure 1 does not show the influenza virus replication strategy at all. It is not clear if authors present the influenza virus proteins or vRNA segments. Figure also does not show how actually mechanism of SARS-CoV-2 replication looks like. There is no clear indication of the template switching mechanism. One of two, the negative sense templates or sg mRNAs (not clear which are presented) are also not shown. In the picture, S and N transcripts are similar in size, while N sgRNA should be shorter, as it has shortest 3’ UTR (also not indicated). Does the black arrow indicate translation of gRNA? Why with use of RTC? The 5’ and 3’ ends should be marked in all RNAs. Authors try to compare different strategies of RNA viruses replication, but not all are presented. What is a conclusion for comparison genomes of different length? The SARS-CoV-2 genome (~30 kb) is much larger that the Influenza virus genome (~13.5 kb).
Comments on Figure 2: Panel A does not show insertion, instead, two pictures in this panel, first and last, present base substitutions, which I believe authors mean by “misalignment”. “misalignment” needs to be changed to “substitution”. Panel B is not clear. Panel C, only m6A is marked, what about other modifications showed in the panel (m5C, C-U editing, and A-I editing)?
The review is difficult to read. I don't feel qualified to judge about the English language and style. However, in my opinion, linguistic proofreading is necessary. Incorrect sentence structure gives a misinterpretation of what is being said. More than once, a source material is necessary to understand the presented information.
Here are some examples:
45-46
“The development of viral titers and clinical symptoms imply that SARS-CoV-2 invades the body for extensive replication and is not effectively detected by host antiviral immunity”
The increase of viral titer?
61-62
Information that influenza virus (IV) has the negative-strand RNA genome is lacking. What is worse, sentences preceding the description of influenza virus replication suggest that IV is one of the positive-sense viruses.
The sentence “(…) influenza viruses, segment and synthesize their genomes for viral replication” is misleading and unclear. Influenza virus does not segment its genome, the virus has segmented vRNA genome.
62-65
Unclear. I do not understand link between replication strategy and broad host range.
69-70
“the transcriptional-regulatory sequence-leader (TRS-L) sequence is located”
“TRS” abbreviation includes word “sequence”, thus it shouldn’t be repeated.
80-81
“Combining DNA nanoball sequencing and nanopore direct RNA sequencing, 7-9 mRNAs can be detected in the cytoplasm of host cells infected by SARS- CoV-2”
It should be clear that “7-9 mRNAs” are viral RNAs, so I propose to introduce “sgRNA” or “sg mRNA” abbreviation for subgenomic mRNA. The abbreviations are introduced in second, third, and fourth chapters what is unlogic.
83
The sentence starts as follows: “this process”, but it is unclear to which process authors are referring to.
98-100
Unclear. Do “distinct domains” mean “distinct regions”? Also, in my understanding, 5’UTR encloses leader sequence, so these are not distinct regions.
100 – 102
Authors should be consistent about nomenclature. The “core sequence “ is TRS.
“The leader sequence comprises approximately 70-90 bases and contains 6-8 nucleotide core sequences”
I believe that this is an information about coronaviruses in general, this is not clear because it follows information presented for SARS-CoV-2. SARS-CoV-2 has 75 nt leader.
103-105
This information is untrue because it is known that the leader sequence contains the SL1 structure, which allows to escape translation termination forced by viral nsp1, thus the leader is a regulatory factor. There is also literature suggesting that the translation initiation of sgRNAs occurs by cap-dependent mechanism. All this information should be discussed.
118
„SARS-CoV-2 generates adaptive evolution”
SARS-CoV-2 undergo evolution or evolve
129-134
“Nucleotide incorporation errors mediated by RdRP mainly exist in the formation of negative strand genomic RNA (-gRNA) and subgenome RNAs (-sgRNAs), followed by the incorporation of erroneous bases into positive strand genomic RNA (+gRNA) and subgenomic mRNAs (+sgRNAs), resulting in “error mutation”[41]. The antiviral drug ribavirin uses this method to mutate RNA virus mutations, thereby reducing viral infectivity[42].”
Unclear and misleading. Ref. 41 is about molnupiravir-induced coronavirus RNA mutagenesis – that is not mentioned in the text. Molnupiravir and ribavirin cause error catastrophe by accumulation of mutations in RNA viruses which already have high mutation rate – this conclusion is not clear from the above description.
What about nsp14 proofreading in case of SARS-CoV-2?
137-140
154-163
Unclear. D614G substitution, with a prevalence of 56%, is an effect of the prominent spike sequence variation, right?
“variegation” mutants?
“viral loading”?
165-169
N501Y is the description for substitution of asparagine to tyrosine, similarly, E484K is substitution of glutamic acid to lysine. Additional explanations are unnecessary.
187-192
330-331
Reviewer 2 Report
The manuscript “Subgenomic RNAs contribute to the rapid duplication of SARS-CoV-2 and COVID19 progression” by Zhang et al. aims to summarize the characteristics and functions of SARS-CoV-2 subgenomic RNAs and their roles in viral replication and evasion of host immune response. The potential of targeting SARS-CoV-2 subgenomic RNAs for drug and vaccine development was also discussed. Although this topic is very interesting and important, this reviewer feels this manuscript is not well organized and there are a lot of aspects need to be improved before considering for publication. Below are my detailed comments:
1. The “Introduction” section better includes some brief description about SARS-CoV-2 the virus itself, such as the viral components and the viral life cycle, in addition to the resulting COVID-19 pandemic. It seems odd without even mentioning what is “subgenomic RNA” in the “Introduction” section of a paper focusing on subgenomic RNA. With that being said, this reviewer suggests moving line 205-215 to the “Introduction” section.
2. It reads relatively weak for section-2 which deals with the generation mechanism of SARS-CoV-2 subgenomic RNAs. Instead, the authors describes more details in section-4 (line 184-194). This adds to the impression that the overall organization of this manuscript could be further improved. This one and my previous comment are two examples. This reviewer suggest the authors carefully think about the organization and content of their manuscript to make it easier for the readers to follow.
3. In all the main sections, the description about subgenomic RNAs fall short. More descriptions were taken by the proteins encoded by the subgenomic RNAs. To this reviewer, it would be great if the authors could try to be more RNA-centered.
4. Since “it is unclear whether the leader sequence can bind to ribosomes in a similar way to IRES…” (line 103-104), this reviewer did not get the point why the authors elaborated on EV71 and IRES (line 92-97). It does not justify the space here if it is only because the authors know a lot about “IRES”.
5. Line 98-99, the leader sequence and the 5´-UTR are overlapping regions if not the same. It is inappropriate to state these as two distinct domains.
6. The authors mentioned the “6-8 nucleotide core sequence” (line 101). What’s the function of the core sequence?
7. In the top panel of Figure 1, the sizes of S-sgRNA and N-sgRNA are a little misleading.
8. “When the RTC synthesizes (-) strand RNA intermediate along (+) strand RNA, it terminates if it encounters the TRS-B sequence and then restarts adjacent to the TRS-L located at the 5´ end of the genome” (line 74-76). This is not always true because otherwise there is only the shortest sgRNA produced since the RTC encounters that TRS-B first. Better to be more cautious with the language here.
9. Line 133, “mutate RNA virus mutations”. Not sure what does this mean.
10. Line 235-236, “The pathway process is to recognize viral surface proteins”. However, “RIG-I-MAVS signaling pathway is considered to be the primary cytoplasmic RNA surveillance and protection system” (line 234-235). What exactly is recognized? RNA or Viral surface proteins?
11. Line 342, “sequence information modification to recognize self-RNA”. Did the author refer this to the 5´-cap? If not, did the author describe this in the manuscript elsewhere except the final “Conclusion” section?
12. Line 55, typo “subgenomic generation”
13. Line 57, please define what is (+) RNA and (-) strand RNA when first appear in the text.
14. Figure 1, please define the red circle (5´-cap) in the figure legend.
15. I found one previously published review paper about SARS-CoV-2 subgenomic RNAs (doi: 10.3390/v13101923). Please cite this paper at appropriate place.
Reviewer 3 Report
This comprehensive review on subgenomic RNAs of SARS-CoV-2 is well written and gives detailed information about the subgenomic generation mechanism, the function of the subgenomic RNAs and their encoded proteins and discusses the subgenomes can be used as potential targets for antiviral drugs.
Minor comments and suggestions:
(1) Figure 1 top panel should show as ORF1a and ORF1b.
(2) Figure 3 is not big enough, the text in the figure is too small.